# TimeSynth: A Framework for Uncovering Systematic Biases in Time Series Forecasting

## Abstract

Time series forecasting is a fundamental tool with wide-ranging applications, yet recent debates question whether complex nonlinear architectures truly outperform simple linear models. Prior claims of dominance of the linear model often stem from benchmarks that lack diverse temporal dynamics and employ biased evaluation protocols. We revisit this debate through **TimeSynth**, a structured framework that emulates key properties of real-world time series—including non-stationarity, periodicity, trends, and phase modulation—by creating synthesized signals whose parameters are derived from real-world time series. Evaluating four model families—Linear, Multi Layer Perceptrons (MLP), Convolutional Neural Networks (CNNs), and Transformers—we find a systematic bias in linear models: they collapse to simple oscillation regardless of signal complexity. Nonlinear models avoid this collapse and gain clear advantages as signal complexity increases. Notably, Transformers and CNN-based models exhibit slightly greater adaptability to complex modulated signals compared to MLPs. Beyond clean forecasting, the framework highlights robustness differences under distribution and noise shifts and removes biases of prior benchmarks by using independent instances for train, test, and validation for each signal family. Collectively, **TimeSynth** provides a principled foundation for understanding when different forecasting approaches succeed or fail, moving beyond oversimplified claims of model equivalence.

## 1 Introduction

Time series forecasting is the task of predicting future values from past observations. It underpins critical decisions in climate prediction, energy planning, finance, healthcare, and transportation (Danese & Kalchschmidt, 2011; Soyiri & Reidpath, 2012; Mounir et al., 2023). Forecasting also involves a set of unique challenges: real-world time series data are non-stationary, display seasonal and periodic patterns, are corrupted by stochastic noise, and undergo abrupt regime shifts (Hu et al., 2024). These factors interact in complex ways, making it difficult to design models that remain both accurate and robust across diverse temporal dynamics.

Time series forecasting has evolved from classical methods such as Autoregressive Integrated Moving Average (ARIMA) to a wide range of deep learning architectures. Transformer-based models, such as Autoformer (Wu et al., 2021) and PatchTST (Nie et al., 2022), introduced mechanisms for long-range dependency modeling, while feedforward approaches like N-BEATS (Neural Basis Expansion Analysis for Time Series) (Oreshkin et al., 2019) and FreTS (Frequency-domain MLP For Time Series model) (Yi et al., 2023) and convolutional networks such as ModernTCN (Modern Temporal Convolutional Network) (Luo & Wang, 2024) and MICN (Multi-scale Isometric Convolutional Network) (Wang et al., 2023) demonstrated alternatives for capturing nonlinear and multi-scale dynamics. More recently, foundation models (Ansari et al., 2024) aim to unify representation learning and forecasting across domains, though their advantages over simpler baselines remain debated.

Recent works have reignited the debate over the relative merits of simple versus complex models for forecasting (Zeng et al., 2023; Li et al., 2023; Toner & Darlow, 2024; Christenson et al., 2024). Surprisingly, single-layer linear networks (Zeng et al., 2023) have often been reported to perform on par with, or even outperform, sophisticated nonlinear architectures. These findings have led to claims that complex model architectures are not essential for time series forecasting (Zeng et al., 2023). However, such conclusions stem from evaluations that lack principled grounding: bench-

marks are dominated by a narrow set of datasets, limited signal diversity (Wu et al., 2021) , and experimental setups that train and test on the same signals or cohorts without adequately controlling for distribution shifts or noise (Forootani & Khosravi, 2025). In the absence of a rigorous framework, it remains unclear whether the observed performance parity reflects true modeling capacity or artifacts of biased evaluation.

Proper evaluation of these claims regarding general applications of forecasting models require datasets whose temporal structure can be systematically varied and experimental conditions carefully controlled. Synthetic datasets offer a promising solution, yet prior efforts have largely relied on arbitrarily chosen parameters that may not reflect the dynamics of real-world signals (Forootani & Khosravi, 2025; Liu et al., 2025). In this work, we introduce TimeSynth, a synthetic framework whose parameters are derived by fitting models to real-life time series signals, ensuring closer alignment with practical forecasting challenges. TimeSynth defines three complementary signal families: (i) *drift harmonic signals*, which combine baseline periodicity with gradual trends; (ii) *phase-modulated signals (SPM-Harmonic)*, which vary frequency and phase to capture modulation effects; and (iii) *dual-component phase-modulated signals (DPM-Harmonic)*, which blend multiple oscillatory modes to emulate richer dynamics. These families are not arranged as a hierarchy of difficulty, but as diverse controlled benchmarks that reveal systematic biases in linear models and highlight contrasting behaviors of nonlinear architectures. Moreover, the proposed framework supports controlled distribution shifts and noise perturbations, enabling proper evaluation of model robustness beyond clean forecasting.

In this paper, we conduct a systematic evaluation of forecasting models across four major families: linear models, multilayer perceptrons (MLPs), convolutional neural networks (CNNs), and transformers. We broaden the evaluation framework from traditional amplitude-based errors (Zhang & Yan, 2023; Nie et al., 2022) to also include metrics that reflect frequency fidelity and phase alignment, providing a complete picture of model behavior. Our results reveal a consistent pattern: linear models underperform across all signal families, not because of signal complexity, but due to a systematic bias. They collapse to a simple oscillations or, at times, the global average, failing to represent the underlying oscillatory structure. In contrast, nonlinear models avoid this collapse by better preserving amplitude, frequency, and phase structure. Among them, transformers and convolutional networks demonstrate greater adaptability to complex modulated signals, while MLPs are comparatively less effective. Beyond clean forecasting, our robustness analysis shows that under controlled distribution shifts, the systematic bias of linear models is prevalent too, whereas nonlinear models retain the flexibility to adjust. Under noise perturbations, robustness depends strongly on the signal family: drift-harmonic signals are more vulnerable to degradation, whereas phase-modulated signals exhibit comparatively stable performance. Finally, we statistically validate these findings using mixed-effects models, confirming the systematic biases and robustness differences across model families. Hence, the key contributions of this paper can be summarized as follows:

- **Principled synthetic framework.** We introduce **TIMESYNTH**, a framework for generating theoretically grounded time-series primitives that encompasses Drift-Harmonic, SPM-Harmonic, and DPM-Harmonic families with precisely controlled shifts, enabling robust model evaluation and revealing linear model bias.

- **Comprehensive evaluation protocol.** We include frequency fidelity and phase alignment along with amplitdue, providing a multifaceted assessment of forecasting performance.

- **Evidence of model biases.** Linear models systematically collapse to a single oscillation or global average, while nonlinear models, especially CNNs and transformers adapt better to complex signals, though no single architecture dominates universally.

## 2 PROPOSED FRAMEWORK: TIMESYNTH

### 2.1 PROBLEM FORMULATION

We focus on the standard setting of univariate time series forecasting. Let $\boldsymbol{x}_{1:T} = \{x_1, x_2, \ldots, x_T\}$ denote a univariate time series of length $T$. At any time step $t$, a forecasting model observes a fixed-length history window of the past $H$ observations, $\boldsymbol{x}_{t-H+1:t} = \{x_{t-H+1}, \ldots, x_t\}$, and is tasked with predicting the next $F$ future values, $\hat{\boldsymbol{y}}_t = \{\hat{x}_{t+1}, \ldots, \hat{x}_{t+F}\}$. The corresponding ground-truth future segment is $\boldsymbol{y}_t = \{x_{t+1}, \ldots, x_{t+F}\}$.

The forecasting objective is to learn a mapping function

$$f_\theta : \mathbb{R}^H \to \mathbb{R}^F, \qquad \text{where } \hat{\boldsymbol{y}}_t = f_\theta(\boldsymbol{x}_{t-H+1:t}), \tag{1}$$

Here, $\theta$ is obtained by minimizing a loss function in forecasting task which is mostly Mean Square Error between $\hat{\boldsymbol{y}}_t$ and $\boldsymbol{y}_t$. $\theta$ is paramterized differently based on the family of forecasting models.

## 2.2 SYNTHETIC DATASET GENERATION

Real-world time series are difficult to evaluate rigorously due to uncontrolled non-stationarity, periodicity, and phase variability. To enable systematic testing, we design three synthetic families—Drift Harmonic, SPM-Harmonic, and DPM Harmonic —deriving parameters from real data to balance realism with controllability. A Graphical Representations of the signals are presented in figure 1

### 2.2.1 DRIFT HARMONIC SIGNAL

The drift harmonic signal extends a pure sinusoidal oscillator by incorporating slow non-stationary trends. It is defined as

$$s(t) = (1 + \epsilon t) \sin(2\pi f t + \phi) + at, \tag{2}$$

where $\epsilon$ controls amplitude drift, $f$ is the base frequency, $\phi$ is the phase offset, and $a$ is a linear trend coefficient. This formulation reflects key properties of many real-world time series: periodicity through the sinusoidal term, phase variability through $\phi$, and non-stationarity via amplitude drift and linear trend. Such dynamics are observed in diverse domains, including circadian rhythms, energy consumption, climate cycles, and engineered oscillatory systems. Parameter ranges are obtained by fitting this model to bandpass-filtered, normalized BVP signals from the PPG-Dalia (Reiss & Schmidt, 2019) dataset using a bounded neural fitting procedure (Bishop & Roach, 1992), with aggregated parameters across subjects to define empirical distributions for synthetic data generation.

### 2.2.2 SINGLE PHASE-MODULATED HARMONIC (SPM-HARMONIC)

The single phase-modulated harmonic introduces a sinusoidal modulation to the phase of the oscillator, enabling local frequency variability. It is defined as

$$s(t) = A \sin(2\pi f t + \beta \sin(2\pi f_{\text{mod}} t)) + \text{offset}, \tag{3}$$

where $A$ is the amplitude, $f$ is the carrier frequency, $\beta$ is the modulation depth, $f_{\text{mod}}$ is the modulation frequency, and the offset aligns the baseline. This design reflects systems where oscillations exhibit timing irregularities, including ECG signals, rotating machinery, and communication signals subject to jitter. The model preserves periodicity while introducing nonlinear variability in instantaneous frequency, providing a controlled way to test forecasting methods under frequency-modulated dynamics. Parameter ranges are derived from ECG segments in the MIT-BIH Arrhythmia Database (Moody & Mark, 2001; Goldberger et al., 2000) using a bounded neural fitting procedure (Bishop & Roach, 1992).

### 2.2.3 DUAL PHASE-MODULATED HARMONIC (DPM-HARMONIC)

The dual phase-modulated harmonic generalizes the SPM-Harmonic by combining two independently modulated oscillatory components. It is defined as

$$s(t) = \sum_{i=1}^{2} A_i \sin(2\pi f_i t + \beta_i \sin(2\pi f_{\text{mod},i} t)) + \text{offset}, \tag{4}$$

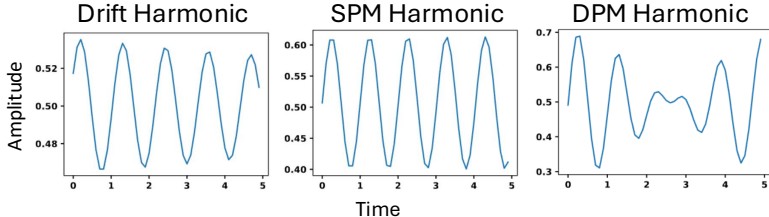

Figure 1: Visualization of 5-second segments from each synthesized signal family.

Table 1: Evaluation metrics used in our study. Each metric includes its definition, formal equation.

| Metric | Definition | Equation |
|---|---|---|
| Amplitude Error | Magnitude fidelity of forecasts, measured with Mean Absolute Error and Mean Square Error. | $\text{MAE} = \frac{1}{H} \sum_{h=1}^{H} \lvert \hat{y}_h - y_h \rvert,$ $\quad\quad$ (5) $\text{MSE} = \frac{1}{H} \sum_{h=1}^{H} (\hat{y}_h - y_h)^2$ $\hat{y}_h$: predicted value, $y_h$: true value, $H$: horizon length. |
| Frequency Error | Spectral fidelity measured by dominant frequency alignment. | $\text{FreqErr} = \lvert \hat{f}_{\text{peak}} - f_{\text{peak}} \rvert \quad (6)$ $\hat{f}_{\text{peak}}$: predicted peak frequency, $f_{\text{peak}}$: true peak frequency. Estimated via FFT with sub-bin interpolation. |
| Phase Error | Temporal alignment via analytic signal–based phase comparison. | $\text{PhaseErr} = \frac{1}{H} \sum_{h=1}^{H} \lvert \hat{\phi}_h - \phi_h \rvert \quad (7)$ $\phi_h$: instantaneous phase of true signal, $\hat{\phi}_h$: predicted phase. Phases extracted from analytic signals $z = y + j\mathcal{H}\{y\}$, $\hat{z} = \hat{y} + j\mathcal{H}\{\hat{y}\}$, with $\mathcal{H}$ the Hilbert transform. |

where $A_i$, $f_i$, $\beta_i$, and $f_{\text{mod},i}$ denote the amplitude, carrier frequency, modulation depth, and modulation frequency of each component, while the offset ensures baseline alignment. Compared to the single-component case, this formulation introduces multi-component periodicity, interacting phase variations, and interference effects, yielding richer nonlinear dynamics. Parameter distributions are derived from ECG segments using the same Database (Moody & Mark, 2001) and the same bounded neural fitting procedure as the SPM-Harmonic. A detailed visualization of all the family of the signals have been provided in Appendix A.2.

## 2.3 TIME SERIES FORECASTING MODELS

For evaluation, we select representative models from four major families that varies the model parameter $\theta$ distinctively. The Linear family includes Linear, DLinear (Zeng et al., 2023), and FITS (Xu et al., 2023), while the MLP family covers 2-layer MLP named MLinear, N-BEATS (Oreshkin et al., 2019), and FreTS (Yi et al., 2023). The CNN family consists of ModernTCN (Luo & Wang, 2024), MICN-Mean, and MICN-Regre (Wang et al., 2023). Finally, the transformer family features a basic Transformer, Autoformer (Wu et al., 2021), and PatchTST (Nie et al., 2022). A detailed descriptions together with hyperparameter configurations are provided in the Appendix A.1.

## 2.4 EVALUATION METRICS

Our evaluation employs three complementary metrics: amplitude error (MAE, MSE), frequency error, and phase error are presented in Table 1. Amplitude error captures the fidelity of predicted signal magnitudes, ensuring that oscillatory envelopes and scaling are preserved. Frequency error evaluates spectral alignment, reflecting whether models maintain the correct oscillatory rates. Phase error quantifies temporal synchronization, which is crucial for signals where even small misalignment translate into large interpretive differences. Together, these metrics provide a comprehensive assessment of forecasting performance beyond amplitude alone.

## 2.5 EVALUATION PARADIGMS

We benchmark models under three experimental conditions: a clean , a noisy and a distribution shift paradigms.

Table 2: Distribution shift setup for the three signal families. Shift 1 and 2 denote lower-frequency ranges, while Shift 3 and 4 denote higher-frequency ranges.

| Family | Shift-1 | Shift-2 | Original (Train) | Shift-3 | Shift-4 |
|---|---|---|---|---|---|
| Drift Harmonic | [0.35, 0.60] | [0.60, 0.85] | [0.85, 1.10] | [1.10, 1.35] | [1.35, 1.60] |
| SPM-Harmonic | [0.00, 0.34] | [0.34, 0.68] | [0.68, 1.41] | [1.41, 2.14] | [2.14, 2.88] |
| DPM-Harmonic | [0.00, 0.34] | [0.34, 0.68] | [0.68, 1.41] | [1.41, 2.14] | [2.14, 2.88] |

### 2.5.1 CLEAN PARADIGM

In the clean setup, models are trained and evaluated on noise-free signals generated randomly using a uniformed distribution within the same parameter ranges. To prevent leakage, we ensure that the training, validation, and test sets consist of signals with distinctive parameters. Specifically, we generate 70 signals for training, 10 for validation, and 20for testing. Each signal spans 300 seconds at a sampling rate of 10 Hz, resulting in sequences of $3,000$ time steps. This protocol guarantees strict separation across splits while maintaining consistent signal characteristics.

### 2.5.2 NOISY PARADIGM

Models are trained exclusively on clean signals but evaluated on noisy counterparts. Noise is injected at test time by adding zero-mean Gaussian perturbations,

$$\tilde{x}(t) = x(t) + \epsilon(t), \qquad \epsilon(t) \sim \mathcal{N}(0, \sigma^2), \tag{8}$$

where the variance $\sigma^2$ is chosen to produce controlled signal-to-noise ratios (SNR). We evaluate across multiple levels, specifically SNR = 40, 30, and 20 dB, to probe model robustness under progressively harsher noise conditions.

### 2.5.3 DISTRIBUTION SHIFT PARADIGM

Models are trained in one parameter range and tested on signals with shifted frequencies, creating a controlled distribution mismatch. This paradigm isolates each model family's ability to extrapolate beyond its training regime.Shift-1 and Shift-4 are the highest in term of shift in opposite direction, where Shift-2 and Shift-3 are more closer to the original range. The amount of shift in frequency is stated in table 2.

## 2.6 STATISTICAL ANALYSIS

We use a linear mixed-effects model (Wiley & Rapp, 2019) to compare all models across clean, noisy, and shifted signals to test the fixed and random effects. This yields robust estimates of overall differences and condition-specific contrasts. The model is defined as follows:

$$\text{Error} \sim C(\text{Model}) \times C(\text{Occasion}) + (1 \mid \text{SignalType}) \tag{9}$$

Here, $C(\text{Model})$ denotes the fixed effect of forecasting model, $C(\text{Occasion})$ denotes the fixed effect of experimental condition (clean, noise levels, or distribution shifts), $C(\text{Model}) \times C(\text{Occasion})$ represents their interaction, and $(1 \mid \text{SignalType})$ specifies a random intercept for each signal family that can be used for fair comparison with the baseline reference. The tests are mainly performed to provide statistical significance to the claims made in the paper.

## 3 PERFORMANCE ANALYSIS

For all experiments, we set the history window, $H$ to 50 steps and the prediction horizon, $F$ to 100 steps. The history length ensured coverage of at least one full cycle for each synthetic signal.

## 3.1 PERFORMANCE ANALYSIS ON CLEAN SETUP

Table 3, demonstrates that on clean signals, nonlinear models consistently achieve lower errors than linear baselines across all signal families and error types, including amplitude, frequency, and phase.

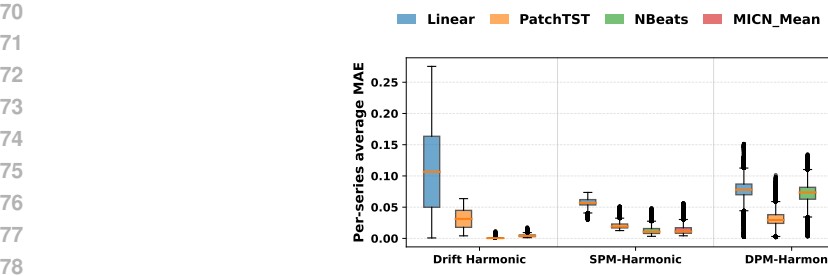

Figure 2: Per-series amplitude error (MAE) on clean Drift Harmonic, SPM-Harmonic, and DPM-Harmonic signals.

While all models capture frequency reasonably well, linear models still lag behind, showing higher errors than their nonlinear counterparts. Within the nonlinear group, CNN- and Transformer-based models achieve superior phase tracking compared to MLP variants, whereas linear models remain systematically poor across all metrics.

Table 3: Mean and median error across amplitude (MAE), frequency, and phase for clean signals. Lower is better. Models are grouped into linear and nonlinear families. This result has been averaged across all three signal families.

| Family | Model | Amplitude (MAE) | | Frequency | | Phase | |
|---|---|---|---|---|---|---|---|
| | | Mean | Median | Mean | Median | Mean | Median |
| **Nonlinear** | MICN_Mean | **0.011** | 0.011 | **0.001** | 0.000 | **6.578** | 6.670 |
| | NBeats | **0.028** | 0.011 | **0.014** | 0.000 | **22.005** | 6.628 |
| | PatchTST | **0.027** | 0.029 | **0.001** | 0.000 | **6.759** | 5.980 |
| | MICN_Regre | 0.030 | 0.011 | 0.023 | 0.000 | 24.918 | 7.043 |
| | FreTS | 0.035 | 0.022 | 0.015 | 0.000 | 27.175 | 13.009 |
| | Transformer | 0.042 | 0.025 | 0.030 | 0.000 | 29.006 | 7.502 |
| | ModernTCN | 0.060 | 0.048 | 0.018 | 0.000 | 26.082 | 8.937 |
| | MLinear | 0.051 | 0.066 | 0.021 | 0.014 | 41.301 | 57.763 |
| | Autoformer | 0.089 | 0.090 | 0.106 | 0.095 | 73.255 | 79.527 |
| **Linear** | DLinear | **0.079** | 0.076 | **0.032** | 0.036 | **54.408** | 50.604 |
| | Linear | **0.081** | 0.078 | **0.036** | 0.041 | **54.998** | 50.085 |
| | FITS | 0.089 | 0.081 | 0.047 | 0.056 | 62.534 | 63.811 |

Figures 2–4b corroborate the trends in Table 3 while providing more information of family specific performance. In amplitude shown in figure 2, linear models fail most on Drift Harmonic due to unmodeled drift, while nonlinear models remain stable, as confirmed by their flat MSE curves over the forecast horizon represented in figure 3. For frequency error depicted in figure 4a, linear models collapse most severely on DPM-Harmonic, reducing forecasts to trivial oscillations, whereas nonlinear models adapt to dominant frequencies.

Phase results in figure 4b reinforce this: linear models ignore phase variation and collapse to trivial oscillations, while nonlinear models actively adjust. Their tighter interquartile ranges demonstrate consistent phase-tracking, and occasional outliers reflect meaningful corrections when tracking rapid phase shifts. CNN- and Transformer-based models (MICN, PatchTST) outperform MLP variants such as NBeats, particularly on complex DPM-Harmonic signals. Collectively, these results show that under clean conditions, linear models oversimplify to trivial oscillations, whereas nonlinear models capture oscillatory structure more faithfully. Comparison within the model family have been shown in Appendix A.3

## 3.2 PERFORMANCE ACROSS DIFFERENT NOISE LEVEL

Robustness to noise varies strongly across signal families. From figure 5a, we can see that Drift Harmonic, with its slowly varying amplitude envelope, is the most vulnerable: additive noise distorts the envelope and induces phase jitter, producing large degradation in phase alignment. By contrast, phase-modulated families (single and dual) maintain near-constant amplitude and stable

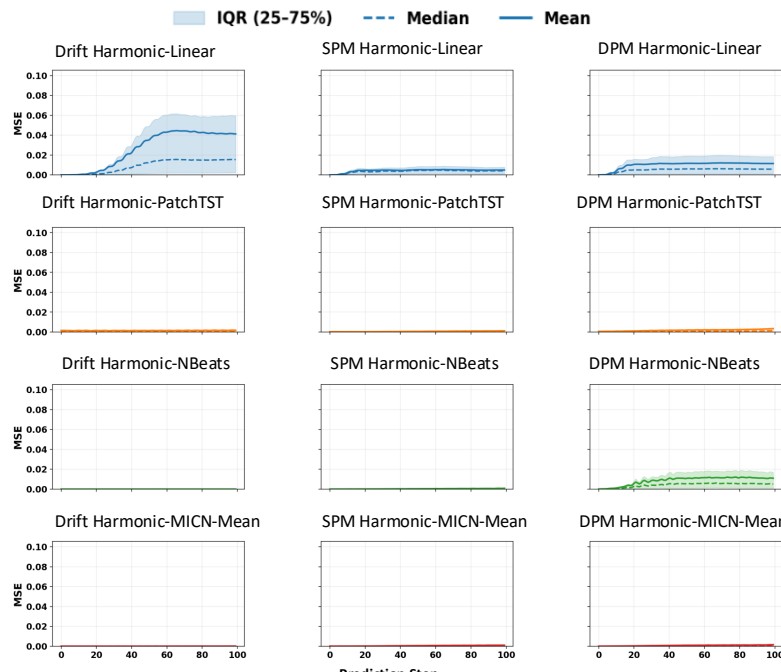

Figure 3: Horizon-wise MSE evolution across prediction steps for linear and nonlinear Models.

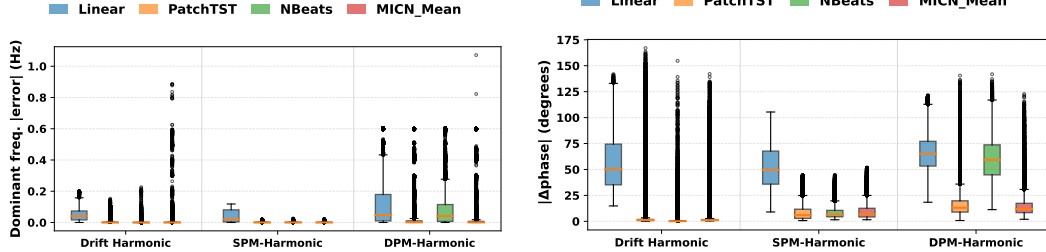

(a) Showing systematic bias in linear models and occasional outliers in nonlinear models due to active correction of oscillations.

(b) showing linear models' collapse to trivial oscillations versus nonlinear models' superior phase-tracking.

Figure 4: Error distributions on clean signals: (a) frequency error, (b) phase error.

oscillatory structure, so noise perturbations remain local and do not accumulate into systematic errors as demonstrated in figure 5b. Furthermore, linear models consistently underperform across all noise levels, indicating that their limitations are structural rather than noise-specific. In Appendix 3.2), a detailed comparison across other signal families is provided. Finally, it should be noted that our experiments only extended up to SNR = 20 dB; at lower SNRs (stronger noise), degradation can become severe across all families.

## 3.3 PERFORMANCE ACROSS DISTRIBUTION SHIFT

When moving to out-of-distribution regimes, we compare models from three complementary perspectives: amplitude, frequency, and phase error. First, amplitude error reveals the strength of nonlinear models in tracking oscillatory envelopes under shift. As shown in figure 6, the MAE for linear models rises sharply in Drift Harmonic signals, while PatchTST and MICN maintain much better amplitude tracking than NBeats. This trend holds consistently across other signal families, with details provided in the Appendix A.5.

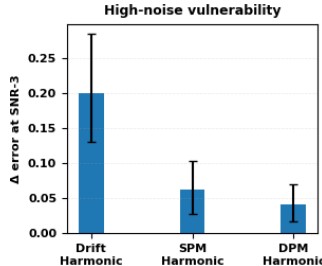

(a) Deviation of phase error showing that noise affects the Drift Harmonic family the most.

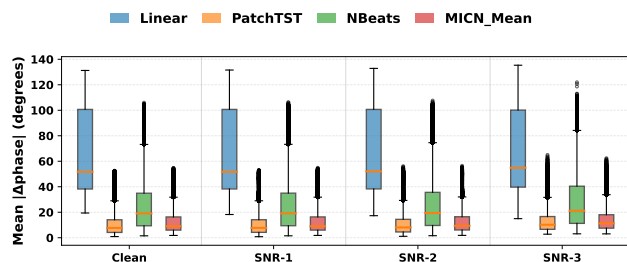

(b) Phase error on noisy SPM-Harmonic signals shows linear models remain consistently poor, with noise having negligible effect.

Figure 5: Phase error under noise conditions: (a) high-noise vulnerability across families, (b) Effect of noise on phase of SPM Harmonic signals.

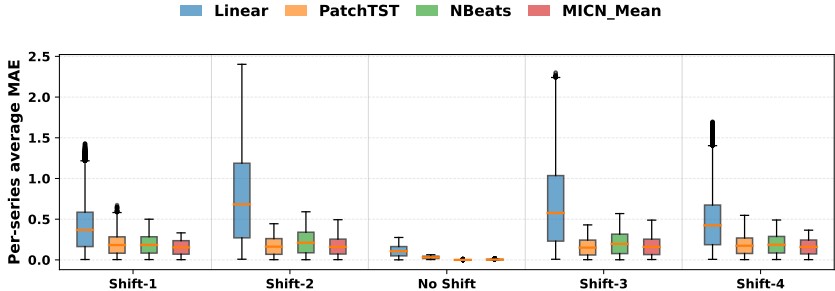

Figure 6: Amplitude error (MAE) under frequency distribution shift for Drift Harmonic.

From the frequency error shown in figure 7, all models degrade as test frequencies move farther from the training range. Nonlinear models perform slightly better than linear in closer shift regions, but the linear family appears deceptively stable across shifts. This stability, however, reflects their collapse to a simple oscillation or global mean, rather than true adaptation to shifted frequencies.

Finally, the phase errors represented in figure 8 complete the picture, where we can see that linear models already incur high phase error in the no-shift setting and remain insensitive across all shifts, which explains their apparent stability. Nonlinear models, by contrast, attempt to track phase changes across shifts, which sometimes succeeds, but also pay the price through large outliers when they fail. In summary, linear models achieve robustness only through collapse to trivial dynamics, whereas nonlinear models expose their sensitivity by adapting to shift, revealing a fundamental trade-off between stability and fidelity. The analysis for other signal families follows a similar pattern and is discussed in detail in the Appendix A.5.

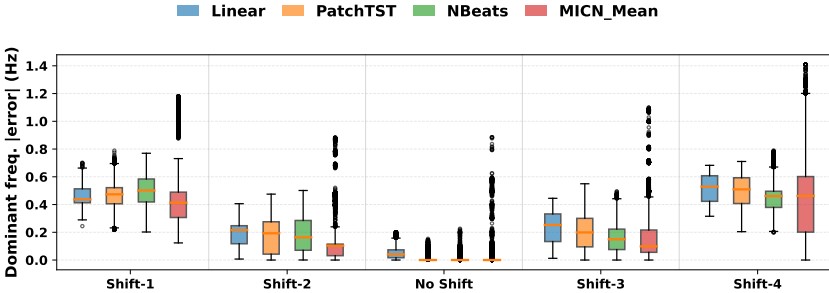

Figure 7: Frequency error under distribution shift across Drift Harmonic, showing degradation with increasing shift. In closer training range the nonlinear model performs better than the linear models.

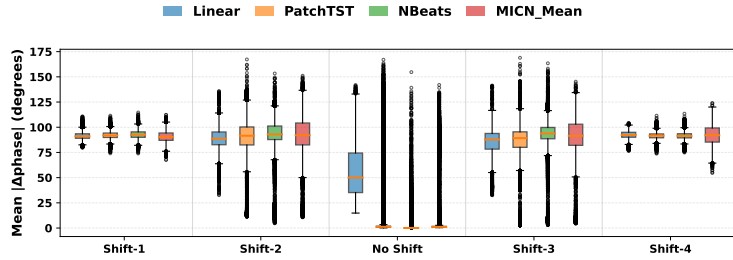

Figure 8: Phase error under distribution shift across Drift Harmonic family, highlighting linear models' insensitivity to phase variation and nonlinear models' adaptive but variable tracking behavior.

### 3.4 STATISTICAL ANALYSIS

Table 4 compares the linear model as baseline against all other models in the clean (no shift) condition and shows that nonlinear architectures achieve significantly lower errors.

Table 4: Mixed-effects contrasts for model performance under clean (No Shift) condition. Values are coefficients (coef) relative to Linear, with corresponding $p$-values. Negative coefficients indicate lower error (better performance).

| Model | Amplitude (coef, $p$) | Phase (coef, $p$) | Frequency (coef, $p$) |
|---|---|---|---|
| Autoformer | +0.009, $p = 0.602$ | +17.03, $p = 0.115$ | +0.174, $p < 0.001$ |
| DLinear | -0.002, $p = 0.913$ | -1.37, $p = 0.899$ | -0.004, $p = 0.817$ |
| FITS | +0.008, $p = 0.646$ | +4.85, $p = 0.654$ | +0.004, $p = 0.832$ |
| FreMLP | -0.046, $p = 0.008$ | -28.46, $p = 0.008$ | -0.029, $p = 0.096$ |
| MICN-Mean | **-0.068, p<0.001** | **-47.87, p<0.001** | **-0.059, p=0.001** |
| MICN-Regre | -0.050, $p = 0.004$ | -31.40, $p = 0.004$ | -0.018, $p = 0.315$ |
| MLinear | -0.030, $p = 0.083$ | -14.06, $p = 0.193$ | -0.018, $p = 0.297$ |
| ModernTCN | -0.022, $p = 0.203$ | -29.13, $p = 0.007$ | -0.028, $p = 0.116$ |
| NBeats | **-0.053, p=0.002** | **-34.75, p=0.001** | -0.033, $p = 0.061$ |
| PatchTST | **-0.053, p=0.002** | **-47.47, p<0.001** | **-0.058, p=0.001** |
| Transformer | -0.040, $p = 0.023$ | -26.63, $p = 0.014$ | -0.011, $p = 0.519$ |

MICN-Mean shows significant improvements in amplitude ($-0.068, p < 0.001$) and phase ($-47.87, p < 0.001$), while PatchTST similarly reduces phase error ($-47.47, p < 0.001$) and frequency error ($-0.058, p = 0.001$). By contrast, the MLP-based NBeats improves less consistently (amplitude $-0.053, p = 0.002$; frequency $-0.033, p = 0.061$). These results confirm the hypothesis that nonlinear models outperform the linear baseline, with MICN and PatchTST providing the strongest advantages.

## 4 CONCLUSION

This work revisited the role of linear and nonlinear models in time series forecasting using **TimeSynth**, a principled framework for synthetic time series data. Across clean, noisy, and distribution-shift conditions, we found a consistent pattern: linear models exhibit a systematic bias, collapsing to simple oscillation or global average regardless of signal family or perturbation. This behavior explains their apparent robustness that they remain unaffected by noise or shift because they fail to represent the underlying oscillatory structure in the first place. Nonlinear models, by contrast, avoid this collapse and demonstrate meaningful adaptation. MLP based model perform adequately on simpler signals, while CNNs and transformers display clear advantages as complexity increases, particularly under DPM-Harmonic Signals. These results highlight that the limitations of linear models are structural rather than condition-specific and that true forecast must be assessed through metrics that capture amplitude, frequency, and phase fidelity. Our framework provides a rigorous foundation for revealing such systematic biases and offers a path toward principled evaluation of when and why different forecasting approaches succeed or fail.

We have added the hyperparameters in the Appendix A.1, for each of the model to reproduce the results. The code for the entire framework will be published upon acceptance.

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

## A    APPENDIX

### A.1    DETAILED HYPER PARAMETER FOR EACH OF THE MODEL

#### A.1.1    LINEAR MODELS

To evaluate linear approaches, we consider three representative models. The first is a basic **Linear** model, serving as the simplest baseline. The second is **DLinear** (Zeng et al., 2023), which decomposes the input into trend and seasonal components before applying a linear transformation. The third is **FITS** (Xu et al., 2023), which builds on the principle that time series can be manipulated through interpolation in the complex frequency domain. The hyperparameters for the linear model have been illustrated in table  5

Table 5: Training hyperparameters for linear-family model

| Hyperparameter | Linear | DLinear | FITS |
|---|---|---|---|
| Training epochs | 200 | 200 | 200 |
| Learning rate | 0.001 | 0.001 | 0.001 |
| Weight decay | 0.01 | 0.01 | 0.01 |
| Cutoff frequency | – | – | 15 Hz |
| Decomposition Kernel Size | – | 15 | – |

#### A.1.2    MLP BASED MODELS

For MLP-based models, we begin with a simple two-layer multilayer perceptron as a baseline called **MLinear**. We then consider **N-BEATS** (Oreshkin et al., 2019), which introduces a deep architecture with backward and forward residual links and a large stack of fully connected layers for expressive temporal modeling. Finally, we evaluate **FreTS** Yi et al. (2023), a frequency-domain MLP that operates in two stages: domain conversion, which maps time-domain signals into complex-valued frequency components, and frequency learning, where redesigned MLPs jointly learn the real and imaginary parts of these components. NBeats is the only model that does backcast as well as forecast. The hyperparameters of MLP models is stated in  6

Table 6: Architectural configurations of MLP-based models, N-BEATS, and FreTS.

| Characteristic | Basic MLP | N-BEATS | FreTS |
|---|---|---|---|
| Number of layers | 2 | 4 per block | 2 |
| Number of blocks | – | 5 | – |
| Hidden units | 256, 512 | 256 per layer | 256 |
| Activation | ReLU | ReLU | ReLU |
| Backcast | No | Yes | No |
| Weight-decay | 0.0001 | 0.001 | 0.0001 |
| learning rate | 0.0001 | 0.0001 | 0.0001 |
| number of epoch | 300. | 5000 | 300 |

### A.1.3 CNN Based Models

For CNN-based models, we evaluate **ModernTCN** (Luo & Wang, 2024) and **MICN** (Wang et al., 2023). ModernTCN is a convolutional architecture designed for time series forecasting, featuring depthwise separable convolutions and residual connections to efficiently capture both short and long-range temporal dependencies. MICN (Multi-scale Isometric Convolution Network) employs a multi-branch structure to capture diverse temporal patterns: local features are extracted through downsampling convolutions, while global dependencies are modeled using isometric convolutions with linear complexity in sequence length. We consider both **MICN-Mean** and **MICN-Regre**, which implement different strategies for handling trend-cyclical components. In Table 7, we have represented the parameters used in both the networks.

Table 7: Architectural and training hyperparameters for CNN-based models.

| Hyperparameter | ModernTCN | MICN |
|---|---|---|
| Number of blocks | $[2, 2, 2, 2]$ | – |
| Large kernel sizes | $[21, 19, 17, 13]$ | – |
| Small kernel sizes | $[3, 3, 3, 3]$ | – |
| Embedding dims | $[64, 128, 256, 512]$ | – |
| Conv kernels | – | $[7, 17]$ |
| Decomposition kernels | – | $[25, 49]$ |
| Isometric kernels | – | $[17, 49]$ |
| Dropout | 0.2–0.4 | – |
| MLP dropout | 0.3 | 0.3 |
| Learning rate | 0.001 | 0.0001 |
| Weight decay | 0.001 | 0.0001 |
| Training epochs | 300 | 300 |
| Patience | 30 | 30 |
| Batch size | 128 | 128 |
| Backcast | No | No |
| Trend Prediction Mode | — | Regression/Mean |

### A.1.4 Transformer Based Models

For Transformer-based models, we evaluate three variants. As a baseline, we include a standard **Transformer** adapted for time series forecasting. We then consider **Autoformer** (Wu et al., 2021), which replaces the self-attention mechanism with an auto-correlation module to better capture long-range dependencies. Finally, we evaluate **PatchTST** (Nie et al., 2022), which processes time series by dividing them into patches and applying Transformer encoders over these patch-level representations. In the Transformer and Autoformer models, half of the history window was used as the label length to warm up the decoder and ensure fair comparison with other models. The hyperparameters of transformer-based models are stated in Table 8

The hyperparameters across different signal families has been largely consistent, with only minor adjustments applied to the learning rate and weight decay.

Table 8: Range of architectural and training hyperparameters for Transformer-based models.

| Hyperparameter | PatchTST | Autoformer | Transformer |
|---|---|---|---|
| Encoder layers | 3 | 2 | 2 |
| Attention heads | 8 | 8 | 8 |
| Embed dimension | 256 | 256 | 256 |
| Feed-forward dim | 256 | 256 | 256 |
| Dropout | 0.2 | 0.2 | 0.2 |
| FC dropout | 0.2 | 0.2 | 0.2 |
| Head dropout | 0.2 | 0.2 | 0.2 |
| Patch length | 15 | – | – |
| Stride | 10 | – | – |
| Label length | – | 25 | 25 |
| Factor | – | – | 3 |
| Training epochs | 2 | 2 | 300 |
| Patience | 80 | 80 | 30 |
| Learning rate | 0.0001 | 0.0001 | 0.0001 |
| Weight decay | 0.0001 | 0.0001 | 0.0001 |
| Batch size | 128 | 128 | 128 |
| RevIN | Yes | Yes | Yes |

## A.2 VISUALIZATION OF SYNTHETIC SIGNALS

Figures 9, 10, and 11 illustrate the three signal families considered in our study. Among them, the DPM-Harmonic signal exhibits the highest complexity, as it combines two distinct frequency components, whereas Drift Harmonic and SPM-Harmonic are governed by single-frequency structures.

## A.3 EXTENDED PERFORMANCE ON CLEAN PARADIGM

### A.3.1 PERFORMANCE ACROSS LINEAR FAMILY

From Figures 12, 13, and 14, we observe that the performance of linear models is nearly the same across all three evaluation metrics. A closer look shows that FITS performs slightly worse than the other linear models on drift harmonic signals, likely due to the amplitude modulation present in the signal.

### A.3.2 PERFORMANCE ACROSS MLP BASED FAMILY

It is evident from Figures 15, 16, and 17 that NBeats outperforms the other models. This advantage stems from the fact that NBeats performs both backcasting and forecasting, giving it stronger reconstruction ability.

### A.3.3 PERFORMANCE ACROSS CNN BASED FAMILY

Both the CNN models perform almost similar in the clean paradigm as seen in 18, 19 20, but MICN does have slightly visible age in amplitude tracking. This may result from employing extract global and local scale simultaneously.

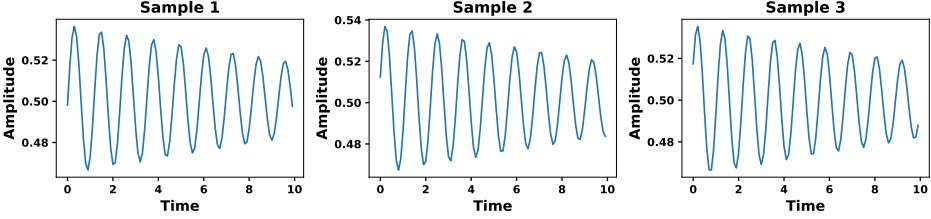

Figure 9: Samples from Drift Harmonic Signal

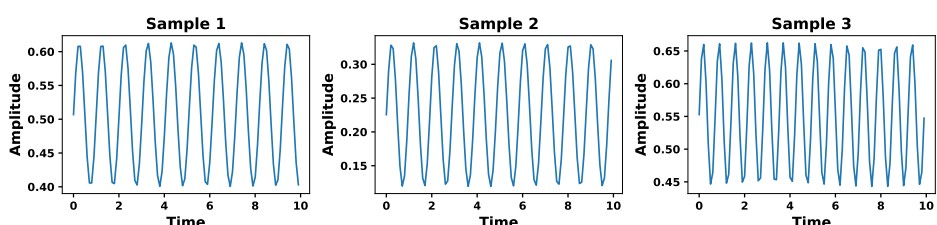

Figure 10: Samples for SPM Harmonic Signal

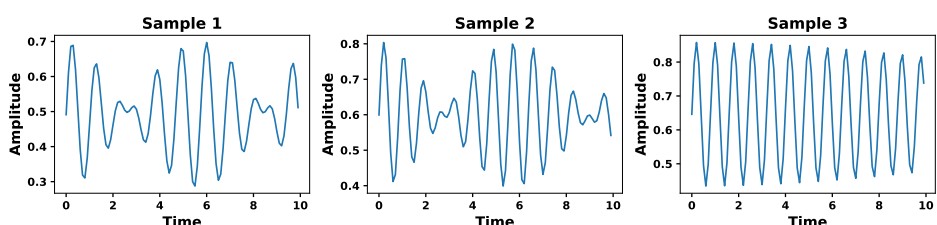

Figure 11: Samples for DPM Harmonic Signal

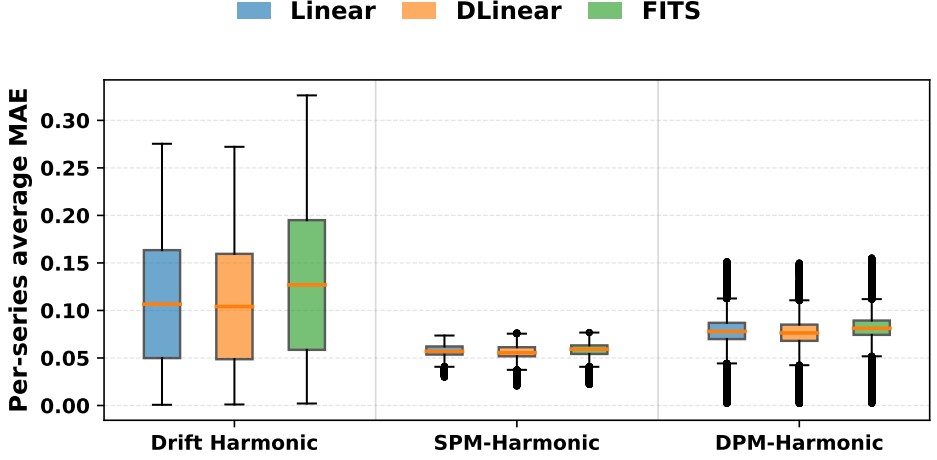

Figure 12: Amplitude Comparison Across Linear Models in Clean Paradigm

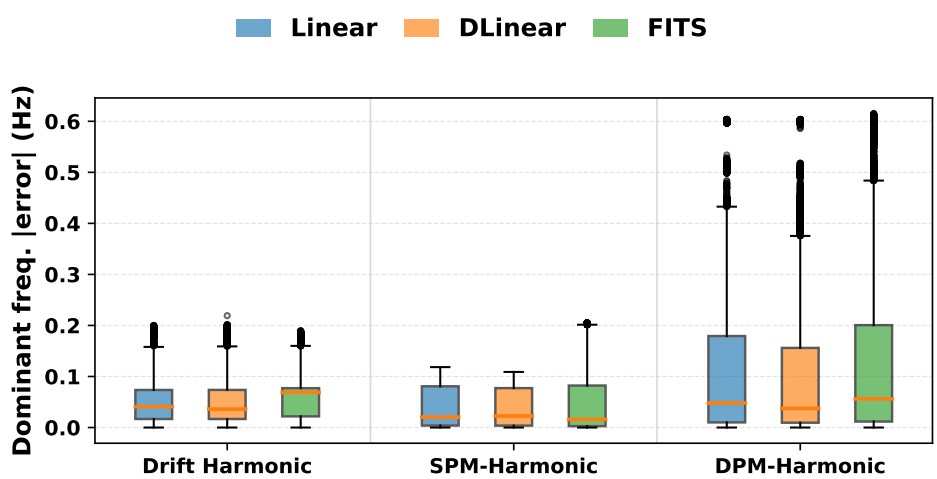

Figure 13: Frequency Comparison Across Linear Models in Clean Paradigm

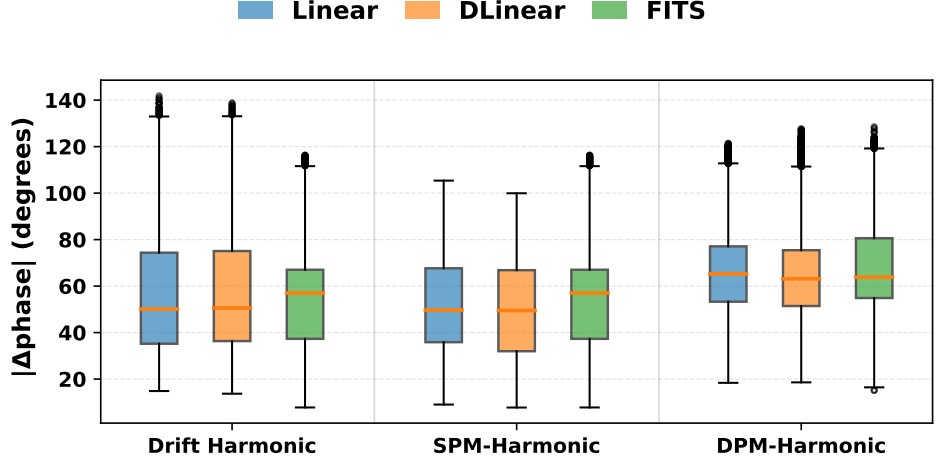

Figure 14: Phase Comparison Across Linear Models in Clean Paradigm

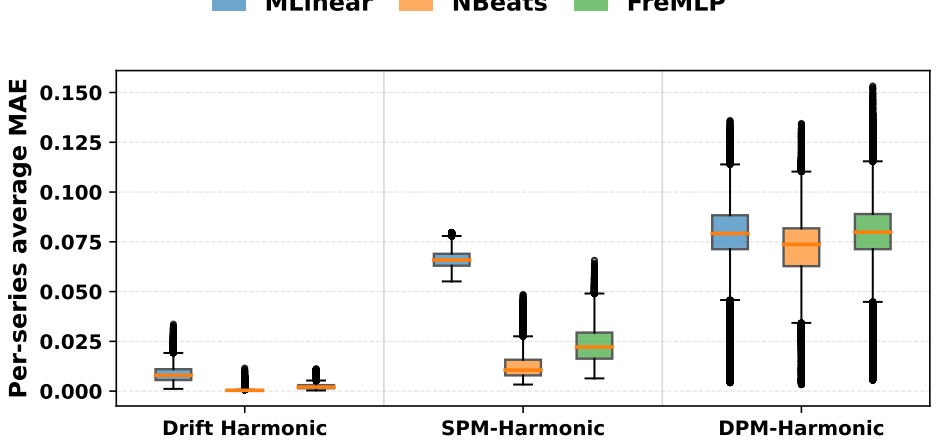

Figure 15: Amplitude Comparison Across MLP-Based Models in Clean Paradigm

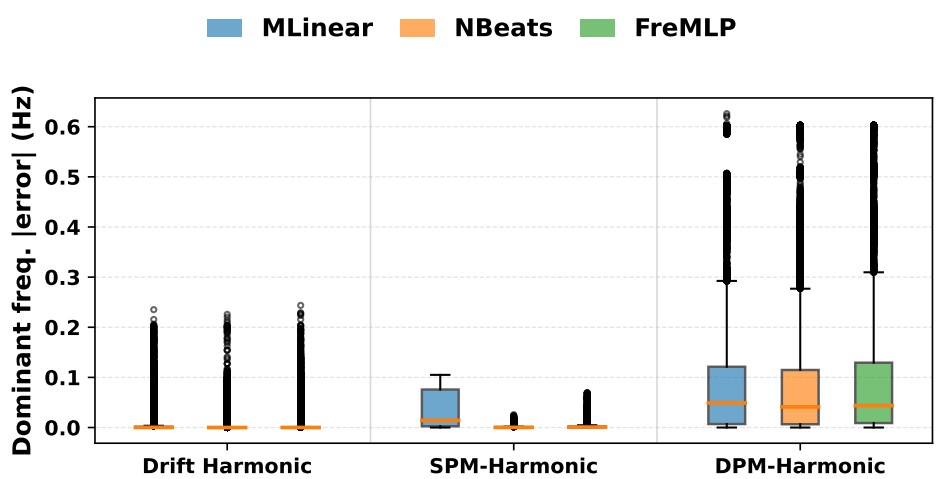

Figure 16: Frequency Comparison Across MLP-Based Models in Clean Paradigm

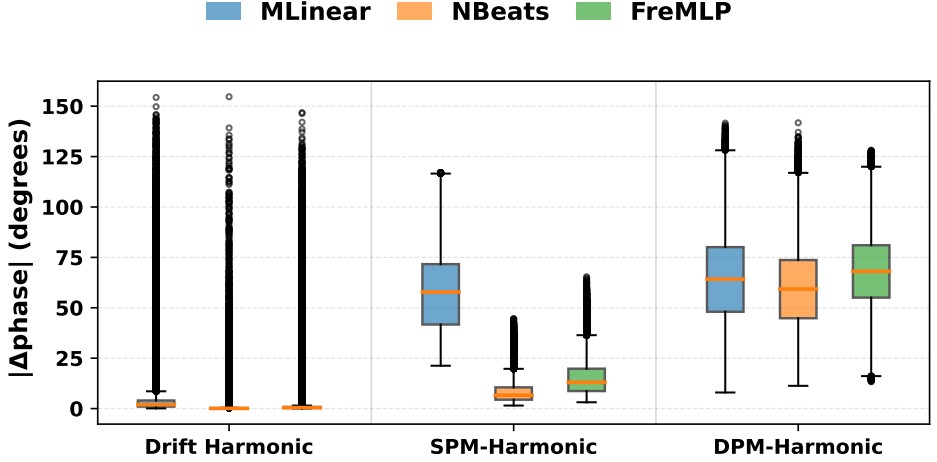

Figure 17: Phase Comparison Across MLP-Based Models in Clean Paradigm

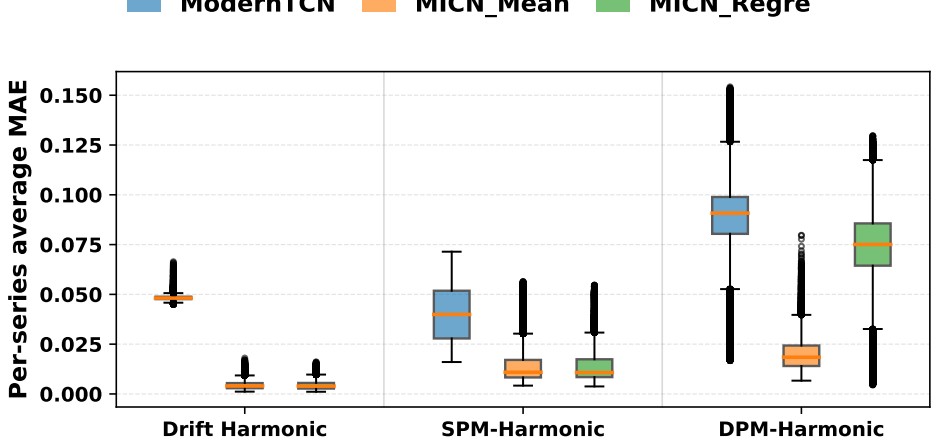

Figure 18: Amplitude Comparison Across CNN-Based Models in Clean Paradigm

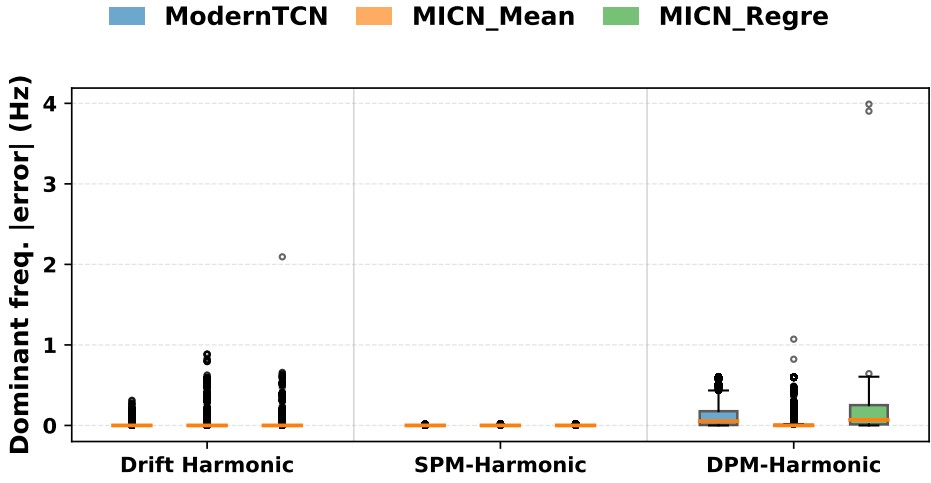

Figure 19: Frequency Comparison Across CNN-Based Models in Clean Paradigm

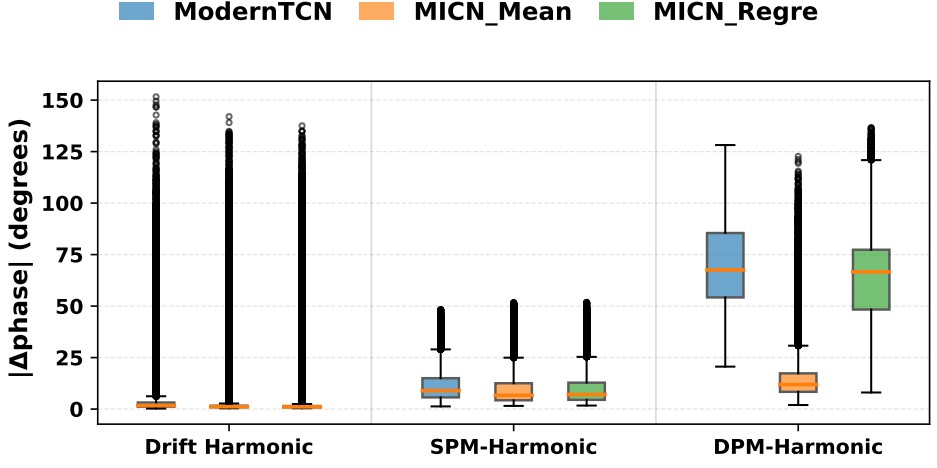

Figure 20: Phase Comparison Across CNN-Based Models in Clean Paradigm

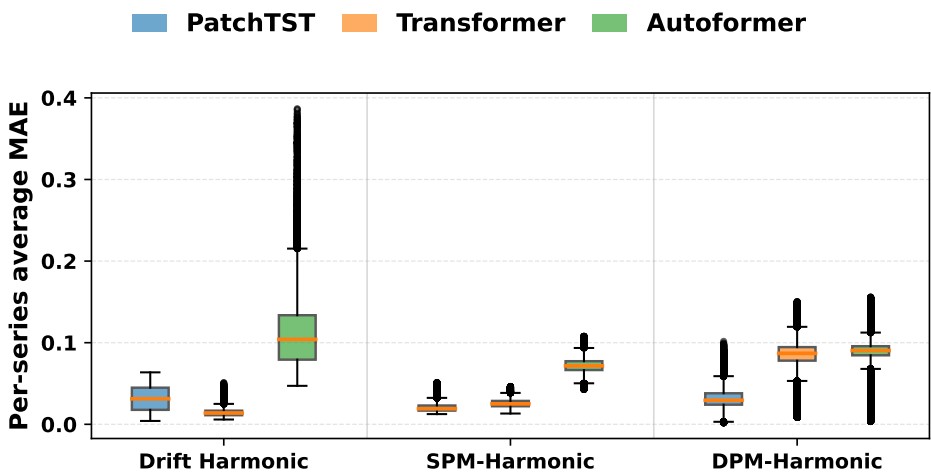

Figure 21: Amplitude Comparison Across Transformer-Based Models in Clean Paradigm

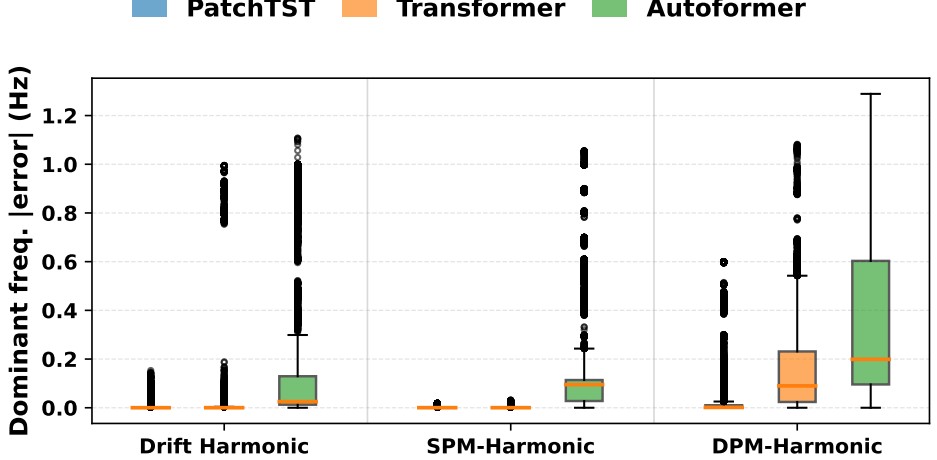

Figure 22: Frequency Comparison Across Transformer-Based Models in Clean Paradigm

### A.3.4    PERFORMANCE ACROSS TRANSFORMER FAMILY

From Figures 21, 22, and 23, we observe that Autoformer underperforms compared to both the Transformer and PatchTST models. This suggests that retaining self-attention is beneficial, as Autoformer replaces self-attention with auto-correlation. Furthermore, PatchTST outperforms the Transformer in certain cases, indicating that patch-based representations can enhance time series forecasting.

### A.3.5    PERFORMANCE ACROSS NONLINEAR MODELS

From figure 3 it is hard to understand the difference between the nonlinear models looking the mse over prediction step and see how the model degrades in the future as compared to other because the nonlinear models were compared against linear models. The error scale of linear model is so high that it makes the error of nonlinear model hard to see. That is the reason we have added a plot in figure 24 showing a comparison amongst the linear model only. That indedd confirms that Convolution based MICN and Transformer based PatchTST performs better than N-Beasts in terms of amplitude prediction over horizon for clean signals.

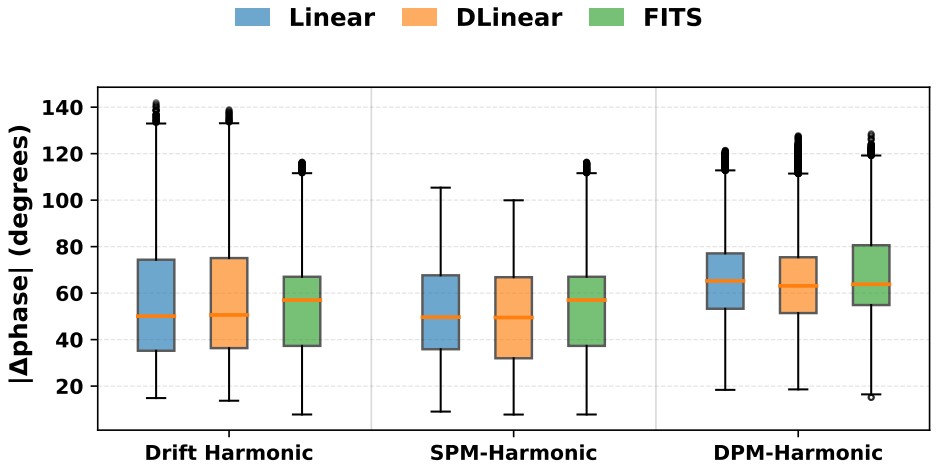

Figure 23: Phase Comparison Across Transformer-Based Models in Clean Paradigm

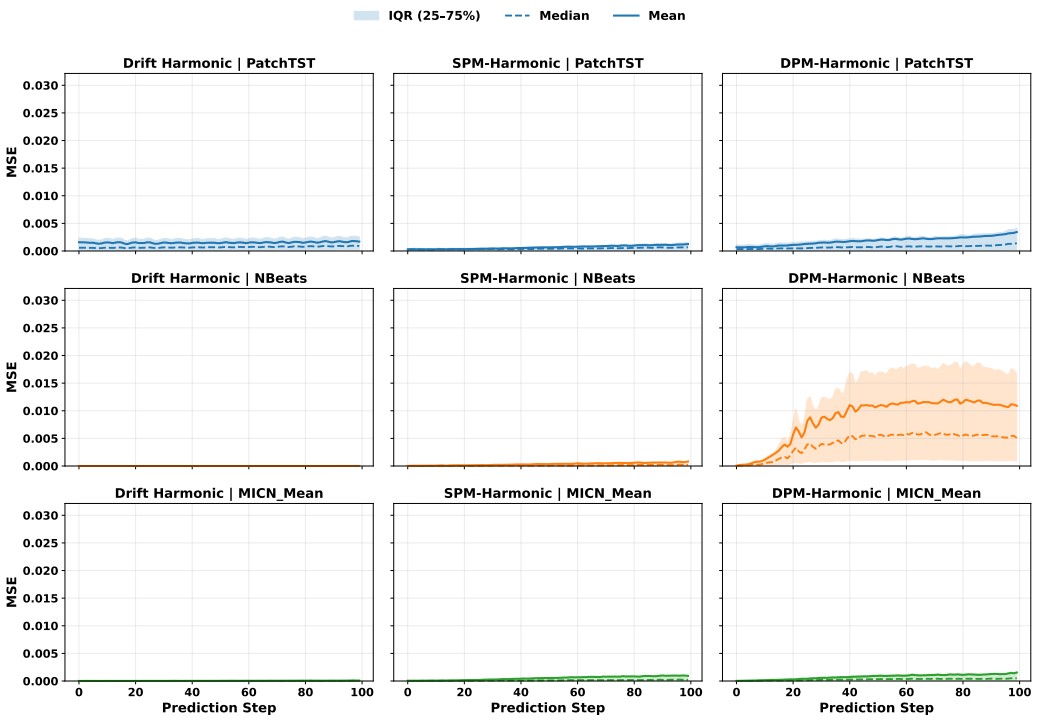

Figure 24: Showing a comparison of MSE over horizon among nonlinear models

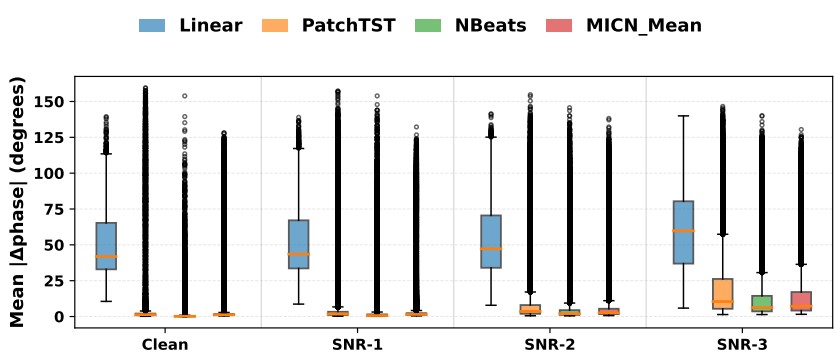

Figure 25: Phase Error Across Drift Harmonic Signal of different forecasting models.

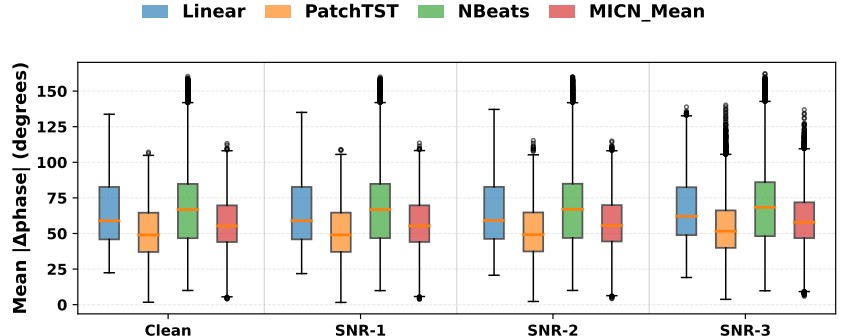

Figure 26: Phase Error Across DPM Harmonic Family of different forecasting models.

## A.4 EXTENDED PERFORMANCE ON NOISY PARADIGM

In this work, we demonstrate that the impact of noise is determined primarily by the signal family rather than the forecasting model. Among the families considered, Drift Harmonic exhibits the greatest vulnerability, while SPM and DPM Harmonic remain comparatively robust. This distinction is evident in the phase error patterns across families, as shown in figures 25, 5b, and 26. The increasing sensitivity of Drift Harmonic stems from its amplitude modulation and underlying trend, where the introduction of noise results in substantial phase distortion relative to the more stable SPM and DPM Harmonic signals.

## A.5 EXTENDED PERFORMANCE OF SHIFT PARADIGM

### A.5.1 EFFECTS ON SPM AND DPM HARMONIC

When examining the MAE across shifts presented in figure 27, 30 for both Drift Harmonic and SPM Harmonic, linear models consistently underperform compared to their nonlinear counterparts. In contrast, the frequency error boxplots displayed in 28 and 31 shows that nonlinear models perform well within the training range (Shift 0) and remain relatively accurate in closer shifts (Shift 2 and Shift 3), but their performance becomes unstable in more distant shifts (Shift 1 and Shift 4). Phase errors in figure 29 and 27 presents a seemingly contradictory picture, where linear models appear more stable. However, this stability is misleading: even at Shift 0, linear models fail to track phase, collapsing to a trivial oscillation that ignores shifts. Nonlinear models, by contrast, attempt to track the underlying phase dynamics, which manifests as wider boxplots and occasional outliers. This pattern reinforces the broader story established in the paper, where Drift Harmonic signals consistently expose the systematic biases of linear models.

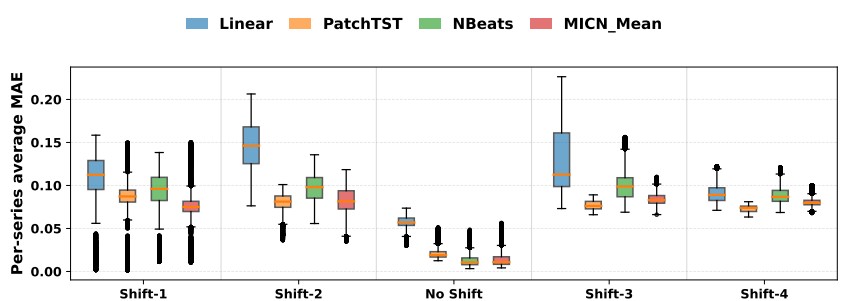

Figure 27: Amplitude Error across different shifted version of SPM Harmonic Signal.

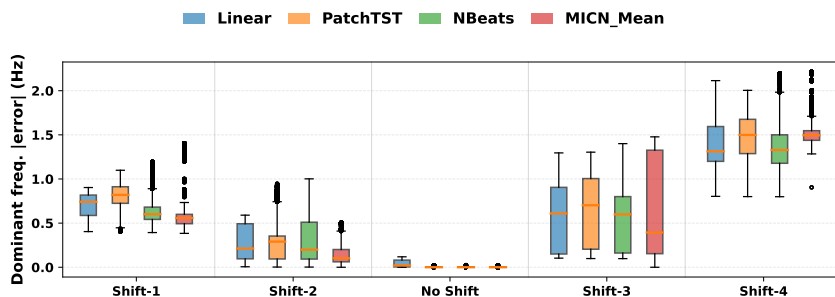

Figure 28: Frequency Error across different shifted version of SPM Harmonic Signal.

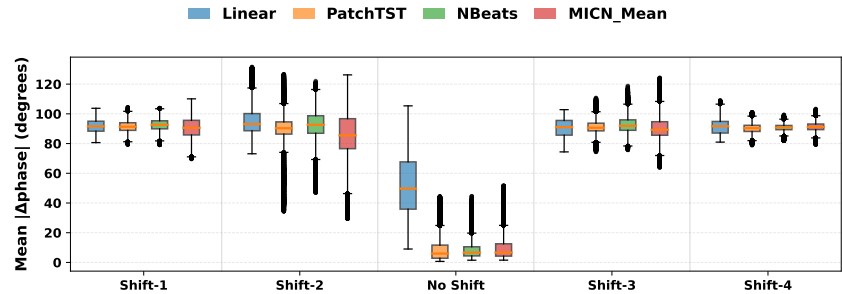

Figure 29: Phase Error across different shifted version of SPM Harmonic Signal.

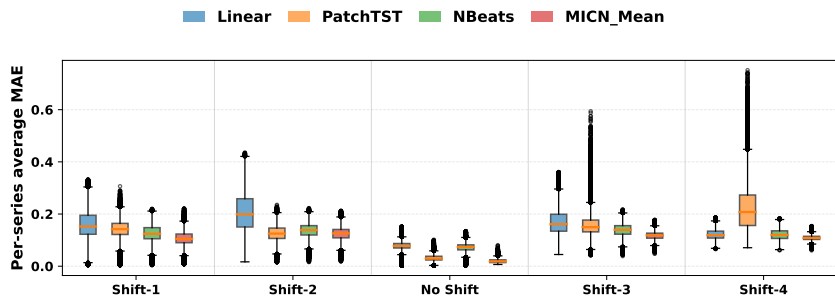

Figure 30: Amplitude Error across different shifted version of DPM Harmonic Signal.

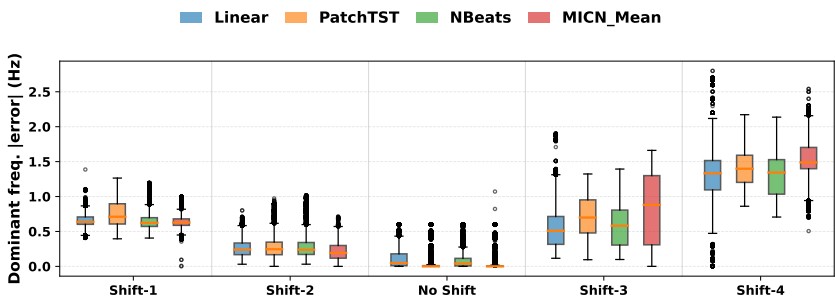

Figure 31: Frequency Error across different shifted version of DPM Harmonic Signal.

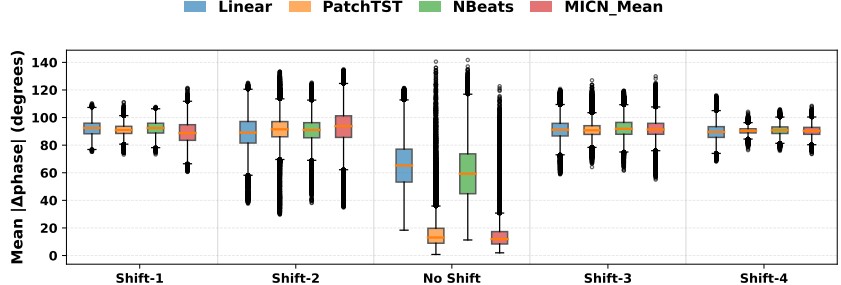

Figure 32: Phase Error across different shifted version of DPM Harmonic Signal.

