# OpenReview forum: "TimeSynth: A Framework for Uncovering Systematic Biases in Time Series Forecasting"
_ICLR.cc/2026/Conference — ICLR 2026 Conference Withdrawn Submission_

### Official Review · Reviewer_T8d7 · 2025-10-25

**Soundness:** 3
**Presentation:** 3
**Contribution:** 3
**Rating:** 2
**Confidence:** 4

**Summary:**

The paper introduces TimeSynth, a synthetic benchmarking framework designed to probe when linear vs. nonlinear forecasting models succeed or fail. Unlike prior synthetic suites with arbitrary parameters, TimeSynth derives parameter distributions from real signals and instantiates three families with controllable complexity and shifts:

Drift-Harmonic: sinusoid + slow trend + amplitude drift

SPM-Harmonic: single Sinusoidal Phase Modulation (frequency/phase jitter)

DPM-Harmonic: dual modulated components (richer interference)

The evaluation goes beyond amplitude errors (MAE/MSE) to include frequency fidelity (peak alignment via FFT) and phase fidelity (Hilbert-phase error). Models from four families are tested: Linear (Linear, DLinear, FITS), MLP (MLinear, N-BEATS, FreTS), CNN (ModernTCN, MICN), and Transformers (Transformer, Autoformer, PatchTST). Across clean, noisy (SNR 40/30/20 dB), and distribution-shift setups (frequency-range shifts), the authors report a consistent phenomenon: linear models “collapse” to trivial oscillations or global means, appearing “stable” under noise/shift only because they fail to track the true oscillatory structure. Nonlinear models—especially MICN and PatchTST—retain frequency/phase structure and degrade more gracefully as complexity increases. A linear mixed-effects analysis supports significance claims.

**Strengths:**

Principled synthetic design: Parameters are fit from real data rather than hand-picked; families capture trend, drift, single/dual modulation, and controlled frequency shifts.

Richer metrics: Frequency and phase errors (plus amplitude) provide a more faithful assessment of oscillatory fidelity than MAE/MSE alone.

Clear empirical finding: Well-documented linear collapse across families and conditions; nonlinear models show superior phase and frequency tracking.

Statistical rigor: Mixed-effects modeling for model × condition interactions, moving beyond single-run anecdotes.

Transparent protocols: Train/val/test split by distinct parameter instances; explicit SNR settings and shift ranges; fixed H/F for comparability.

**Weaknesses:**

Domain narrowness of parameter derivation. Real-parameter fitting is taken from physiological signals (PPG-DaLiA, MIT-BIH ECG). Claims about general time-series forecasting would be stronger if parameters were also derived from non-physio domains (load, economics, climate).

Univariate only. TimeSynth is univariate; many real settings are multivariate with cross-lag structure. Conclusions may not carry over.

Evaluation choices need tightening for DPM. Peak-frequency error is ambiguous for dual-component spectra; matching/assignment for multiple peaks isn’t fully specified (could penalize the “right” peak).

Phase metric robustness. Hilbert-phase can be fragile near low amplitude and under noise; unwrapping and failure-mode handling aren’t fully detailed (e.g., discarded frames).

Classical baselines missing. No ARIMA / ETS / state-space (SSM/Kalman) comparison; these are canonical linear baselines distinct from “linear layers.”

Limited robustness axes. Distribution shift manipulates frequency only; real shifts also change drift (a), modulation depth (β), or amplitude-drift (ε). Noise is white Gaussian only (no colored noise or outliers).

Capacity / parity questions. It’s unclear whether parameter counts, receptive fields, normalization (e.g., RevIN), and tuning budgets were balanced across model families.

Sample size & sensitivity. Only 70/10/20 series per family; results may be sensitive to corpus size and to (H, F) choices (fixed at H=50, F=100).

**Questions:**

DPM evaluation: How do you match predicted vs. true multiple peaks? Please specify the assignment rule (e.g., top-k with Hungarian matching) and report top-2 peak errors.

Phase computation: How do you handle phase unwrapping and amplitude-near-zero frames in the Hilbert transform? Provide robustness checks and fraction of stabilized/filtered frames.

Baselines: Can you add ARIMA and a simple state-space (SSM) baseline with your frequency/phase metrics?

Generalization of parameter sources: Do results hold when parameter distributions are derived from non-physiological datasets (load, macro, climate)?

H/F sensitivity: Provide a grid over H ∈ {24, 96, 168} and F ∈ {24, 96, 336} to confirm that the linear-vs-nonlinear gap is not an artifact of context/horizon choices.

Shift realism: Add shifts on β, a, ε and evaluate. Also evaluate colored noise (pink/AR(1)) and outliers.

Fairness table: Report per-model params, FLOPs/step, receptive field, normalization/de-trend, tuning budget, and multi-seed confidence intervals.

---

### Official Review · Reviewer_M7jH · 2025-10-28

**Soundness:** 2
**Presentation:** 2
**Contribution:** 3
**Rating:** 4
**Confidence:** 4

**Summary:**

This paper presents TimeSynth, a synthetic benchmark framework for systematically evaluating time series forecasting models. The authors introduce three families of signals—Drift Harmonic, Single Phase-Modulated Harmonic (SPM-Harmonic), and Dual Phase-Modulated Harmonic (DPM-Harmonic)—with parameters derived from real-world physiological data. They evaluate a wide array of linear, MLP, CNN, and transformer-based forecasting models, comparing amplitude, frequency, and phase fidelity under clean, noisy, and distribution-shifted scenarios. The study uncovers a structural bias in linear models: they consistently regress to trivial oscillation irrespective of signal complexity, whereas nonlinear models (notably transformers and CNNs) better adapt to challenging signal dynamics.

**Strengths:**

**S1** The paper introduces a synthetic benchmark (TimeSynth) whose parameterization is carefully grounded in fits to real-world physiological time series, providing a rigorous and controlled testbed for forecasting method evaluation. The drift, phase modulation, and dual-modulation primitives are well-motivated and visually illustrated in Figure 1 and Appendix figures (e.g., Figures 9–11).

**S2** Evaluation is more holistic than usual, spanning amplitude (MAE/MSE), frequency alignment, and phase error metrics (see Table 1), thus probing forecasting beyond naïve point error.

**S3** The authors execute a thorough empirical study across multiple paradigms (clean, noise, distribution shift) with robust experimental design. For example, test and train splits use non-overlapping parameterizations, which addresses a common “data leakage” critique in time series evaluation.

**Weaknesses:**

**W1** Limited Novelty of Core Idea: While the framework is more principled than prior synthetic benchmarks, the design is essentially an aggregator of improved synthetic signal families and systematic model evaluation. The creation of synthetic time series for benchmarking and the classic “linear vs. nonlinear” debate are both very well-trodden. The novelty is more in the comprehensive execution and the systematic bias claim, but less in core methodological innovation.

**W2** Framework Coverage Limited to Univariate Signals: All evaluations are on synthetic, univariate signals. No analysis of multivariate time series or cross-channel temporal relations is included; this constrains the generalizability of the claims, especially as many prominent forecasting methods are motivated as multivariate solutions.

**W3** Real-World Generalizability Questionable: All signal families, though parameterized by fits to real signals, are fundamentally stylized and miss complex patterns (irregularities, regime changes, missing data, warping, etc.) seen in actual applications. While synthetic signals are justified for controlled benchmarking, the authors should explicitly address scenario mismatches that arise outside this synthetic regime.

**W4** Lack of Theoretical Depth: The paper’s “systematic bias” claim for linear models is only empirically supported. There is no accompanying analytical or theoretical discussion of why the observed collapse to trivial oscillation happens (e.g., spectral or optimization perspective). The evaluation is thorough experimentally, but a rigorous mathematical treatment of the phenomenon is missing. An explicit analysis of the learned weights/spectra of the linear models would be insightful and is notably absent.

**Questions:**

**Q1** Can the authors provide a deeper theoretical or empirical rationale for the “systematic collapse” phenomenon observed in linear models, possibly by analyzing learned parameter spectra, weight patterns, or optimization trajectories? What specific characteristics of TimeSynth signals trigger this failure mode?

**Q2** Why are classic nonlinear (but non-deep) models (e.g., threshold autoregressive, Markov-switching, kernel methods) missing from the baseline pool, given their relevance to the linear vs. nonlinear debate?

**Q3** Will the authors extend TimeSynth to multivariate signals, or incorporate real-world irregularities (e.g., nonstationary noise, missing data, regime switching) in future work?

**Q4** Can the authors clarify the axis scaling/interquartile labeling in key figures (especially horizon-wise error plots), and describe error bar construction?

---

### Official Review · Reviewer_KpGB · 2025-10-30

**Soundness:** 2
**Presentation:** 2
**Contribution:** 2
**Rating:** 2
**Confidence:** 4

**Summary:**

This paper introduces TimeSynth, a principled synthetic framework to probe when linear vs. nonlinear forecasters succeed or fail. It derives parameter distributions from real signals, then instantiates three controllable families—Drift-Harmonic (trend + amplitude drift), SPM-Harmonic (single phase modulation), and DPM-Harmonic (two modulated components)—and evaluates four model families (linear, MLP, CNN, Transformer) across clean, noisy (Gaussian SNR), and distribution-shift (frequency-range shifts) regimes. Metrics go beyond amplitude (MAE/MSE) to include frequency peak error and phase error (Hilbert transform). Mixed-effects analyses support the main claim: linear models exhibit a systematic collapse to trivial oscillation/mean, while nonlinear models maintain amplitude–frequency–phase structure and are more adaptable to modulation and shifts.

**Strengths:**

The three signal families isolate non-stationarity, modulation, and multi-component structure with parameters sourced from real datasets, enabling targeted stress tests rather than ad-hoc toy signals. Adding frequency and phase fidelity provides a more faithful view of oscillatory forecasting than amplitude-only scores; plots (horizon-wise) help interpret failure modes. Clean vs. noise (SNR 40/30/20 dB) vs. distribution shift (five disjoint frequency ranges) show when models truly generalize vs. appear “robust” by collapsing.

**Weaknesses:**

1. The framework restricts to univariate forecasting; many real applications hinge on multivariate dependencies (cross-channel/lag interactions). Results and claims may not transfer.
2. “Bounded neural fitting” for deriving parameter distributions (PPG-Dalia, MIT-BIH) lacks details (objective, bounds, priors, initialization, regularization, failure handling), making replication and bias assessment difficult.
3. The linear family’s “collapse” may partly reflect optimization/regularization/normalization choices (e.g., detrending, mean removal, frequency-domain linear extrapolators). It’s unclear that stronger linear spectral baselines (Fourier extrapolation, seasonal naive, STL+AR) were tuned or included beyond DLinear/FITS.
4. Add sensitivity studies for frequency/phase metrics: FFT window length, tapering, sub-bin interpolation method; phase unwrapping strategy; robustness on multi-peak spectra (use spectral-earth-mover distance or multi-peak matching as a sanity check).

**Questions:**

As in Weaknesses.

---

### Official Review · Reviewer_8zG1 · 2025-11-02

**Soundness:** 2
**Presentation:** 2
**Contribution:** 3
**Rating:** 4
**Confidence:** 4

**Summary:**

This paper revisits the ongoing debate regarding the comparative performance of simple linear models versus complex nonlinear architectures (like CNNs and Transformers) for time series forecasting. The primary objective is to investigate whether prior claims of linear models' dominance are artifacts of flawed benchmarks.

To address this, the authors introduce TimeSynth, a synthetic data generation framework. This framework generates three families of signals: Drift Harmonic, Single Phase-Modulated (SPM) Harmonic, and Dual Phase-Modulated (DPM) Harmonic . A key aspect of this methodology is that the signal parameters are not arbitrary but are derived from fitting models to real-world physiological data (PPG and ECG signals).

The authors evaluate four model families (Linear, MLP, CNN, and Transformer)  using a comprehensive set of evaluation metrics. Critically, this evaluation extends beyond typical amplitude-based errors (MAE/MSE) to include frequency fidelity and phase alignment errors. The experimental setup is divided into three paradigms: a clean setup, a noisy setup (testing robustness to additive Gaussian noise), and a distribution shift setup (testing extrapolation to unseen frequencies).

The paper's key finding is that linear models exhibit a systematic bias, consistently "collapse to simple oscillation" or a global average, thereby failing to capture the underlying signal structure. This collapse occurs regardless of the evaluation paradigm. The authors conclude that the perceived robustness of linear models under noise and distribution shift is an illusion; they appear "robust" only because they fail to model the signal's dynamics in the first place. In contrast, nonlinear models, particularly CNNs and Transformers, avoid this collapse and demonstrate superior performance in capturing amplitude, frequency, and phase, especially on more complex modulated signals.

**Strengths:**

- The work directly engages with a critical, unresolved debate in the field. By shifting the focus from which model is "best" to why models fail, it provides a much-needed mechanistic explanation for performance discrepancies.

- The TimeSynth framework uses parameters derived from real-world physiological signals (PPG, ECG) rather than arbitrary ones. Its design of three signal families (Drift, SPM, DPM) allows for a controlled assessment of model capabilities against specific dynamic properties.

- The methodology is extended beyond standard amplitude-based errors (MAE/MSE) to include frequency and phase error metrics. The authors argue these are essential for capturing certain model failure modes, and the results (e.g., Figure 4) are used to illustrate this point.

**Weaknesses:**

- The TimeSynth framework is parameterized exclusively from BVP (PPG) and ECG signals. Both are physiological time series known for their strong, quasi-periodic, harmonic nature. The paper's framing and conclusions, however, speak to "real-world time series" in general, which include domains like finance, energy, and climate  that are dominated by very different dynamics (e.g., stochastic trends, chaotic behavior, sharp, non-periodic shocks). The current framework, being entirely harmonic-based, may not be representative of these other domains.

- The paper's claims about the superiority of Transformers in complex scenarios  would be more convincing if tested against more recent, state-of-the-art Transformer-based architectures, as the field has advanced significantly.

- The paper omits any discussion or comparison with the emerging class of time series foundation models. Given that the paper cites Chronos, its absence from the experimental comparison is a notable gap, as these models are highly relevant to the paper's core debate.

- The paper makes broad claims about families of models (Linear, CNN, MLP, Transformer) based on the performance of a few selected exemplars (e.g., MICN-Mean for CNNs, PatchTST for Transformers) . This approach risks model-specific artifacts, where the observed behavior may be unique to the chosen model rather than the entire family. The conclusions would be more generalizable if they were based on an aggregated or averaged performance across a wider, more diverse set of models within each family.

- The analysis of the results lacks depth in some areas. For instance, many box plots (e.g., Figure 4b, Figure 8, Figure 29) display a significant number of outliers, particularly for nonlinear models. The paper notes these as "occasional outliers reflect[ing] meaningful corrections" or "large outliers when they fail"  but does not provide a sufficient discussion or deeper insight into the specific conditions or signal properties that cause these extreme failure modes. This is a missed opportunity for a more nuanced analysis.

- The y-axis scale in Figure 3 renders all results for nonlinear models completely illegible, obscuring a central finding. This plot should be corrected in the main paper, for example, by using a log scale or separate y-limits.

- (Minor issue) The large file size of the document significantly impedes loading and review.

- (Minor issue) Using the lowercase 'figure' instead of the standard "Figure" for references (e.g., "figure 5a" ) should be corrected.

**Questions:**

Please refer to the Weaknesses.

---

### Note · Authors · 2025-11-18

I have read and agree with the venue's withdrawal policy on behalf of myself and my co-authors.